# Predicting Wafer-Level Package Reliability Life Using Mixed Supervised and Unsupervised Machine Learning Algorithms

**DOI:** 10.3390/ma15113897

**Published:** 2022-05-30

**Authors:** Qing-Hua Su, Kuo-Ning Chiang

**Affiliations:** Department of Power Mechanical Engineering, National Tsing Hua University, Hsinchu 300, Taiwan; 0967356474shq@gmail.com

**Keywords:** Wafer-Level Package (WLP), Finite Element Analysis (FEA), machine learning, Kernel Ridge Regression (KRR), Cluster algorithm

## Abstract

With the increasing demand for electronic products, the electronic package gradually developed toward miniaturization and high density. The most significant advantage of the Wafer-Level Package (WLP) is that it can effectively reduce the volume and footprint area of the package. An important issue in the design of WLP is how to quickly and accurately predict the reliability life under the accelerated thermal cycling test (ATCT). If the simulation approach is not adopted, it usually takes several ACTCs to design a WLP, and each ACTC will take several months to get the reliability life results, which increases development time considerably. However, simulation results may differ depending on the designer’s domain knowledge, ability, and experience. This shortcoming can be overcome with artificial intelligence (AI). In this study, finite element analysis (FEA) is combined with machine learning algorithms, e.g., Kernel Ridge Regression (KRR), to create an AI model for predicting the reliability life of electronic packaging. Kernel Ridge Regression (KRR) combined with the K-means cluster algorithm provides a highly accurate and efficient way to obtain AI models for large-scale data sets.

## 1. Introduction

Reliability is an important topic in the field of electronic packaging. Solder ball reliability analysis is the key to measuring the reliability of WLP. One main cause of package failure is thermal-induced CTE (coefficient of thermal expansion) mismatch between different materials. In ATCT, the first solder ball failure usually occurs at the diagonal corner of the package; this is the location with the largest distance from the neutral point (DNP). Although traditional ATCT tests can obtain the reliability life result, they are too time-consuming (usually several months). The long experiment time leads to decreased R&D efficiency, which cannot meet the market demand.

Finite element simulation is a feasible approach to reducing design cycles and time. Lin et al. [1] built a finite element model for WLP. The reliability life can be obtained by substituting incremental equivalent plastic strain into the Coffin–Manson model, and simulation and experiment results are in good agreement. Because the element mesh size in the upper right corner of the solder ball is crucial to the final simulation result, Cheng [2] built a 3D finite element model for an area array type package. In this study, we will build our simulation database for machine learning by following our lab modeling experiences [1,2,3,4].

For FEA, different researchers may lead to different results. Moreover, it still takes time (several days or weeks) to get the simulation results. Therefore, it is necessary to introduce machine learning to lower the training threshold of simulation, unify the results, and reduce development time. Machine learning can be divided into supervised and unsupervised learning based on the presence or absence of artificially assigned labels. Among the two algorithms involved in this study, kernel ridge regression (KRR) is supervised learning, and cluster analysis is unsupervised learning. This study uses the K-means algorithm for cluster analysis. Arthur and Robert [5] first proposed the idea of ridge regression (RR) to solve the multicollinearity of the data dimension. With the evolutionary progress of the algorithm, Kernel Ridge Regression (KRR) was proposed in 2000 by Cristianini and Shawe-Taylor [6]. The essence of KRR is the combination of RR and kernel tricks. K-means clustering was proposed by Macqueen [7]. It can perform data partitioning to reduce the workload of KRR; this would significantly reduce the training CPU time of the prediction model.

On the other hand, many researchers have used different types of algorithms to effectively predict the reliability of solder balls, such as Artificial Neural Network (ANN) [8], Support Vector Regression (SVR), Random Forest (RF) [9], and so on. In this study, we compare KRR, and KRR with K-means (K-K) with other algorithms in performance error and training CPU time.

## 2. Fundamental Theory

### 2.1. Reliability Life Prediction Model

The reliability life prediction method of the packaging structure can be mainly divided into two types: the energy-based method and the strain-based method. The Coffin–Manson model [10] used in this research belongs to the strain-based method. The incremental equivalent plastic strain is the key to evaluating the solder’s reliability life. The expression of the Coffin–Manson model is shown in Equation (1).
(1)$$N_{f}=\alpha {\left(\Delta \varepsilon_{eq}^{pl}\right)}^{\phi }$$

$N_{f}$ is the reliability prediction life (cycle), and $\Delta \varepsilon_{eq}^{pl}$ is the incremental equivalent plastic strain. α and ϕ are empirical constants. In this study, α and ϕ are 0.235 and −1.75.

### 2.2. Ridge Regression (RR)

We often use the Least Squares Method (LSM) to solve regression problems in statistics. It is a mathematical optimization modeling method that finds the best function match of the data by minimizing the sum of squares of the error. The squared loss function of LSM is shown as Equation (2).
(2)$$L\left(\beta \right)={\left(y-X\beta \right)}^{T}\left(y-X\beta \right)$$

In Equation (2), X is the matrix expression of the data input. The row of the matrix is the number of data samples. The column of the matrix is the data dimension. y is the output. β is the equation coefficient. In LSM, our target value $\hat{\beta }$ is the value that minimizes L(β). To do this, we need to solve the partial derivative of L(β) to β. The resulting expression of $\hat{\beta }$ is shown as Equation (3).
(3)$$\hat{\beta }={\left(X^{T}X\right)}^{-1}X^{T}y$$

Training models using LSM enables them to fit known sample points quickly and accurately, but it may cause overfitting. In Equation (3), when there is multicollinearity in the data dimension, $X^{T}X$ is no longer a full-rank matrix, and it would be challenging to solve its inverse matrix directly. RR is an algorithm proposed to solve this problem. RR is essentially an improved LSM. By giving up the unbiasedness of the LSM, a more realistic mathematical model is obtained at the expense of losing some information and reducing accuracy. Equations (4) and (5) show the new loss function and target value expression.
(4)$$J\left(\beta \right)={\left(y-X\beta \right)}^{T}\left(y-X\beta \right)+k{\left\|\beta \right\|}^{2}$$
(5)$$\beta ={\left(X^{T}X+kI\right)}^{-1}X^{T}y$$

In Equations (4) and (5), k is the ridge parameter, and I is a matrix of normal numbers. As k increases, the undetermined coefficient β would stabilize, and we were looking for the smallest k value under the condition that the coefficient is stable.

### 2.3. Kernel Ridge Regression (KRR)

KRR [11,12,13] combines RR and kernel tricks. In many cases, it requires mapping data into high-dimensional space to improve machine learning performance. It is found that the same effect can be achieved directly by defining a function *K*. This function *K* is called the kernel function. There are three commonly used kernel functions: polynomial kernel, sigmoid kernel, and radial basis function (RBF) kernel, shown as Equations (6)–(8). As we can see, there are three parameters in the polynomial kernel. The sigmoid kernel has two parameters; the RBF kernel only has one parameter, which is its strength. The amount of calculation of KRR is significantly reduced by utilizing the RBF kernel. In this study, we focus on the RBF kernel.
(6)$$K\left(x_{i},x_{j}\right)={\left(\gamma \langle x_{i},x_{j}\rangle +b\right)}^{d}$$
(7)$$K\left(x_{i},x_{j}\right)=tanh\left(\gamma \langle x_{i},x_{j}\rangle +b\right)$$
(8)$$K\left(x_{i},x_{j}\right)=exp\left(-\gamma \|x_{i}-x_{j}\|{}^{2}\right)$$

In Equations (6)–(8), $x_{i}$ and $x_{j}$ represent two data, 〈·,·〉 represents dot product, and $\left\|x_{i}-x_{j}\right\|$ is the Euclidean distance between $x_{i}$ and $x_{j}$.

We need to write the RR solution as an inner product to introduce the kernel function. Converting the original formula to a particular form requires the matrix inversion lemma [14].

Consider a general partitioning matrix $M=\left(\begin{array}{cc} E & F\\ G & H \end{array}\right)$. Assuming that both E and H are invertible, we have:(9)$${\left(E-FH^{-1}G\right)}^{-1}=E^{-1}+E^{-1}F{\left(H-GE^{-1}F\right)}^{-1}GE^{-1}$$
(10)$${\left(E-FH^{-1}G\right)}^{-1}FH^{-1}=E^{-1}F{\left(H-GE^{-1}F\right)}^{-1}$$
(11)$$\left|E-FH^{-1}G\right|=\left|H-GE^{-1}F\right|\left|H^{-1}\right|\left|E\right|$$

We use Equation (10) to simplify the optimal solution of β. Let $H^{-1}≜k^{-1}I$, F$≜X^{T}$, G≜−X, E≜I, then we obtain a new expression shown as Equation (12).
(12)$$\beta ={\left(X^{T}X+kI\right)}^{-1}X^{T}y=X^{T}{\left(kI+X^{T}X\right)}^{-1}y$$

Now, it is very close to kernelization. Our remaining task is to predict $y^{*}$ when a new sample point $x^{*}$ comes in. We write β in the form of vector summation, and let $\alpha ≜{\left(kI+X^{T}X\right)}^{-1}y$. Then, we rewrite Equation (12) to Equation (13).
(13)$$\beta =X^{T}\alpha =\overset{N}{\underset{i=1}{\sum }}\alpha_{i}x_{i}$$

We can find that β is just a weighted average of all samples. Thus, the predicted value for a new sample is:(14)$$y^{*}=\beta^{T}x^{*}=\overset{N}{\underset{i=1}{\sum }}\alpha_{i}x_{i}{}^{T}x^{*}=\overset{N}{\underset{i=1}{\sum }}\alpha_{i}K\left(x^{*},x_{i}\right)$$

The predicted value is the weighted average of the inner product of the new sample and all the old samples. After converting the original formula to the inner product form, we selected different kernel functions to simplify our calculations.

### 2.4. K-Means Clustering

K-means [15] is an algorithm that implements cluster analysis based on the principle of minimum distance. The K value must be given in advance, representing the number of cluster centers. For each iteration, we need to calculate the mean of the sample points in the cluster to update the cluster center. K-means clustering divides the *n* samples into *k* sets so that the within-cluster sum of squares (WCSS) is the smallest. The formula we use is shown in Equation (15). The updated formula for cluster centers is shown in Equation (16).
(15)$$E=\sum_{i=1}^{k}\sum_{p\in C_{i}}dist{\left(p,c_{i}\right)}^{2}$$
(16)$$m_{i}\left(c_{i}\right)=\sum_{j=1}^{n}p_{j}/n$$

In Equation (15), $c_{i}$ is the cluster center, p is one sample point, and $dist\left(p,c_{i}\right)$ is the Euclidean distance from p to the cluster center.

## 3. WLP FEA Model Validation

It is assumed that the CTE difference between the substrate and the wafer is Δα. The DNP of the solder ball is L. After selecting material parameters, Δα is fixed. Solder balls farther from the chip’s center have a greater impact on deformation mismatch due to thermal loading. In the case of WLP, the thermal loading failure usually occurs at the outermost diagonal solder ball of the package. This study uses five WLP test vehicles (TV: WLP1-5) [16,17,18] and one fan-out WLP (FO-WLP, [19]) for FEA model validation. The structure component sizes, materials, and mean-time-to-failure (MTTF) reliability life are shown in Table 1, Table 2 and Table 3 [17,18,19]. This research uses these data to verify our simulation results. In order to reduce the computational time cost, this study adopts a simplified two-dimensional finite element model and sets the following basic assumptions: each structure is with homogeneous and isotropic materials; the temperature of the structure is uniform; residual stress is not considered; and all the contact surface between different materials is considered as perfect bonding.

Considering that the package body is a symmetric structure, semi-diagonal two-dimensional models were used in this study to simplify the modeling and finite element analysis processes; examples can be seen in Figure 1, Figure 2, Figure 3 and Figure 4, and PLANE 182 has been selected as the element type in ANSYS (Figure 4). The model boundary condition is fixed for all nodes in the center of the structure in the x-direction. The node at the bottom of the center of the structure is fixed in the x- and y-direction to avoid rigid body motion.

The FEA model for WLP 1-5 includes the following materials: silicon chip; low-k layer; stress buffer layer (SBL); redistribution layer (RDL); solder ball; under bump metallurgy (UBM); copper pad; printed circuit board (PCB); and solder mask. In addition, to further simplify the 2D model, the actual model simplifies the connection between UBM and the solder ball. The element mesh size in the upper-right corner of the solder ball would affect the final simulation result. Based on our previous research experience, the mesh size of this key position is fixed; it is located on the upper-right corner of the outmost solder ball. The controlled mesh size in height and width is 7.5 μm and 12.5 μm, and is shown in Figure 4.

Table 3 shows linear elastic material parameters for the WLP model. They are Young’s Modulus €, Poisson’s Ratio (*ν*), and CTE. For the solder ball, we use the Chaboche Kinematic Hardening model (Equation 17) to fit the nonlinear behavior of the solder at different temperatures, and the obtained fitting parameter table is shown in Table 4. The stress–strain curve for solder balls in different temperatures in this study is shown in Figure 5 [20].
(17)$$\alpha_{}=\frac{C_{}}{\gamma_{}}\left(1-e^{-\gamma_{}\varepsilon^{pl}}\right)+\sigma_{0}$$
where *α* is the back stress, *σ*_0_ is initial yield stress, *C* is constant for proportional to hardening modulus, *γ* is the rate of decrease of hardening modulus, and Δ*ε**^pl^* is increment plastic strain, individually.

Finally, the thermal cycling load [22,23,24,25,26] was applied to our FEA model according to the JEDEC JESD22-A104D Condition G, with a temperature range of −40 ℃ to 125 ℃. We fixed the ramp rate. It is 16.5  ℃/min, and the dwell time is 10 min. The total time for a complete temperature cycle is 40 min. The thermal cycling temperature profile is shown in Figure 6.

After eight cycles, the incremental equivalent plastic strain will be stabilized, and it can be input to the Coffin–Manson equation to calculate the reliability prediction life of WLP. The reliability life between experiment and simulation is shown in Table 5. We can see that the difference between experiment and simulation prediction reliability life falls within an acceptable range for five test vehicles. Therefore, the WLP simulation models can be trusted. This study uses a validated simulation procedure and a controlled mesh size, as well as an automatic model generation technique we developed to create a large database of different design parameters and use it for machine learning.

## 4. Machine Learning Prediction Results

### 4.1. Supervised Learning—KRR

In this study, we select four design parameters that greatly influence the reliability of WLP in building the database. They are chip thickness (CT), SBL thickness (SBLT), upper pad diameter (UPD), and lower pad diameter (LPD). Other structure parameters are fixed as WLP-2. The diagram of structure design parameters is shown in Figure 7.

The design parameters for different training sets are shown in Table 6, Table 7, Table 8, Table 9, Table 10 and Table 11.

In Table 11, we randomly pick 100 data as testing data. Five training datasets (Table 6, Table 7, Table 8, Table 9 and Table 10) are used to train the KRR model separately and use the testing dataset to test the model’s generalization. This study applies grid search for parameter optimization, and the data preprocessing is MinMaxScaler. The best model performance for each training set is shown in Table 12.

We have four criteria for measuring the quality of the model. They are maximum training difference, average training difference, maximum testing difference, and average testing difference. The model’s generalizability is the main concern of this research, and the testing performance of the model is shown in Figure 8 and Figure 9.

From Figure 8, we can see that the maximum testing difference is very stable; it shows that our trained model will not overfit. From Figure 9, we can see that the average testing difference gradually decreases with the increase of training data; this means the accuracy of the model is increasing. Therefore, we can continue to add training data to improve the model performance. On the other hand, judging from the growth curve of the training CPU time, one can conclude that using a larger dataset would cause the model to take a long time to train. In order to reduce the training time of the model, this study introduces the K-means algorithm.

### 4.2. Supervised/Unsupervised Machine Learning—KRR Mixed with K-Means

First, to demonstrate the effectiveness of K-means, a larger training dataset should be generated. This research mixes five training datasets (Table 6, Table 7, Table 8, Table 9 and Table 10) with 1296 testing data (Table 11), and removes duplicate data; the final total data is equal to 9601. We randomly pick 9000 data as training data, and 601 as testing data. When the pure KRR algorithm is used for training, the total time spent is 1340 s. The specific performance of the model is shown in Table 13.

In Table 13, we can see that the model’s prediction accuracy is further improved. The average training and testing differences are both under five cycles. Our target now is to reduce the model training time. Using K-means as the preprocessing step of KRR, the training data is divided into K clusters. Each cluster corresponds to a sub-model with KRR. When we input the testing data, the test data uses a K-means model to determine which cluster it belongs to, and is trained on that particular sub-model. With K = 4, we show the performance of the K-K model in Table 14 and Table 15.

In Table 14, “n” represents the number of training data in each cluster; the total data number is 9000. “m” represents the number of testing data in each cluster; the total data number is 601. The final maximum difference is the maximum value among all sub-models. The average training difference is the weighted average. The final training and testing results are shown in Table 15, and the accuracy is similar to the pure KRR model. In this study, the focus of the K-K model is the training CPU time. In Table 16 and Figure 10, it can be seen that as the value of K increases, the training time decreases rapidly.

From Figure 10, we can find that the CPU time gradually decreases, and the trend slows down as the value of K increases. The average difference remains stable; both training and testing are under five cycles. In addition, a comparison of different machine learning algorithms using 9000 training data and 601 testing data is presented in Table 17.

In Table 17, the performance of the KRR model and the SVR model is very close in terms of training time and training error. The K-K hybrid model achieves similar accuracy as the pure KRR, ANN, and SVR algorithms. However, in terms of training time, it can reach the level of the RF algorithm, which dramatically improves the training efficiency of the algorithm.

## 5. Conclusions

This study, using validated FEA, created five training datasets and one testing dataset for machine learning. In comparison with FEA, the results indicated that the KRR and the K-K machine learning algorithms are fast and effective in predicting the reliability life of WLP. With the increase of training data, the accuracy of the AI model is gradually improved. However, as the AI dataset grows, training time will increase dramatically, making it necessary to reduce the training time. Using a hybrid model combining K-means and KRR can significantly reduce training time while maintaining similar prediction accuracy. When K is 32, we can obtain a data prediction model with an average error of around four cycles, and the overall CPU training time is 7 s, much less than the pure KRR model’s 1340 s training time. As compared with ANN, RF, and SVR, the K-K model is undoubtedly fast and accurate.

## Figures and Tables

**Figure 1 materials-15-03897-f001:**
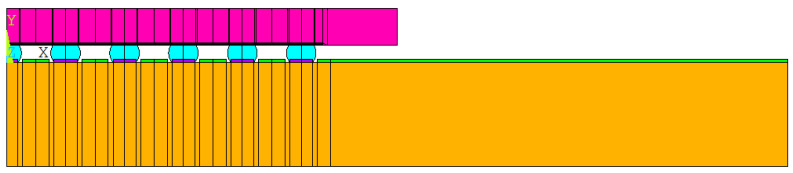
WLP-1 semi-diagonal FEA model.

**Figure 2 materials-15-03897-f002:**
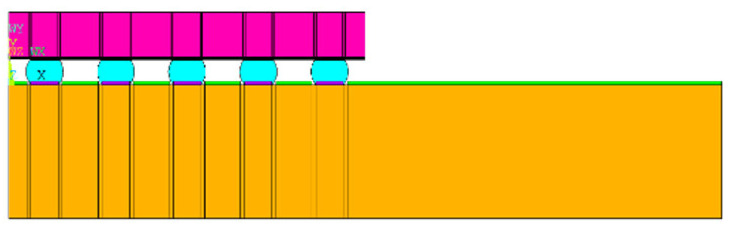
WLP-2 semi-diagonal FEA model.

**Figure 3 materials-15-03897-f003:**
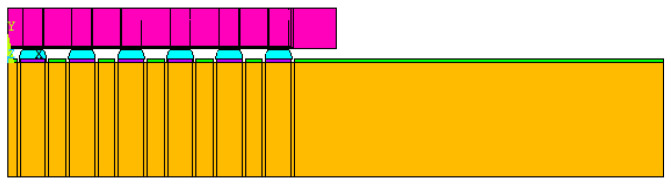
WLP-3 semi-diagonal FEA model.

**Figure 4 materials-15-03897-f004:**
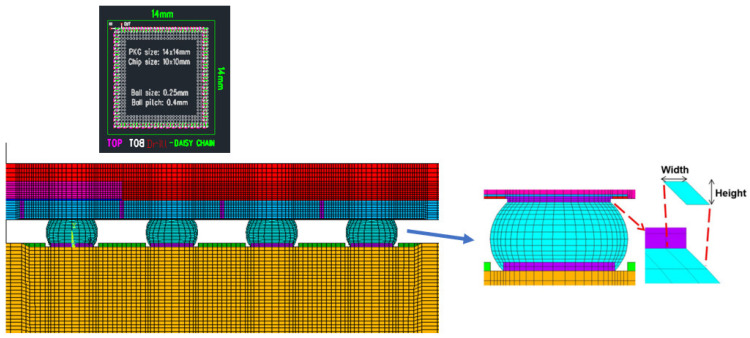
FO-WLP and the schematic of critical mesh size.

**Figure 5 materials-15-03897-f005:**
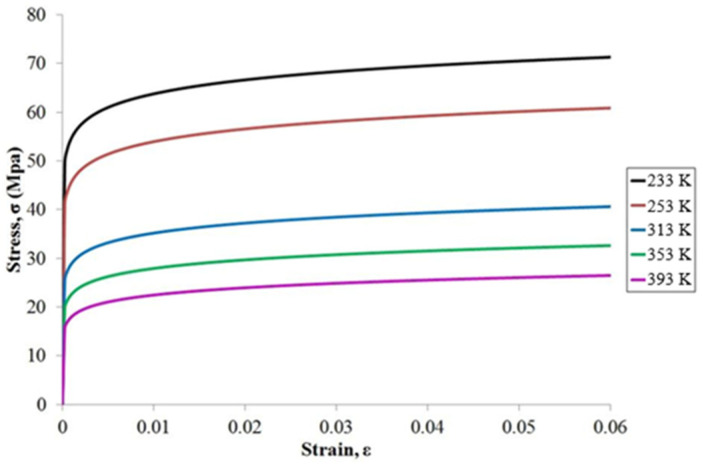
The stress–strain curve for SAC305 [19,21].

**Figure 6 materials-15-03897-f006:**
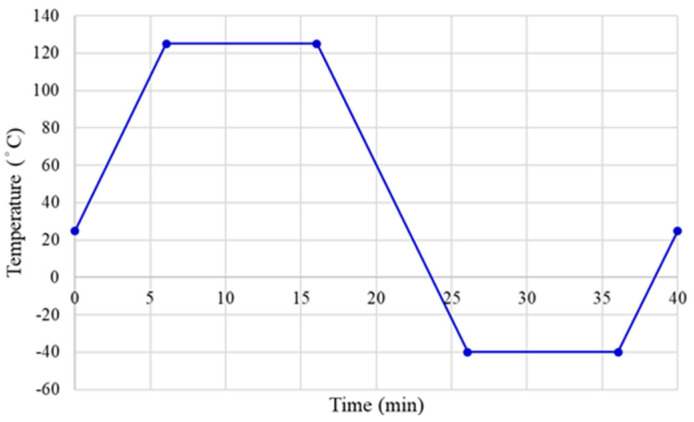
Thermal cycling temperature profile.

**Figure 7 materials-15-03897-f007:**
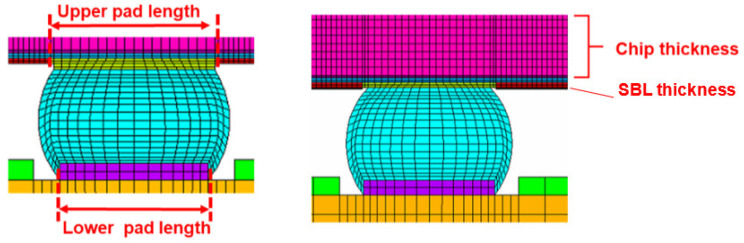
The diagram of WLP design parameters.

**Figure 8 materials-15-03897-f008:**
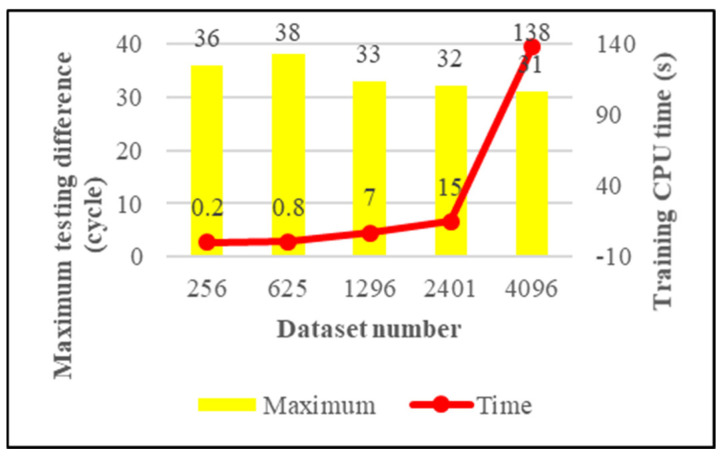
The maximum testing difference of the KRR model.

**Figure 9 materials-15-03897-f009:**
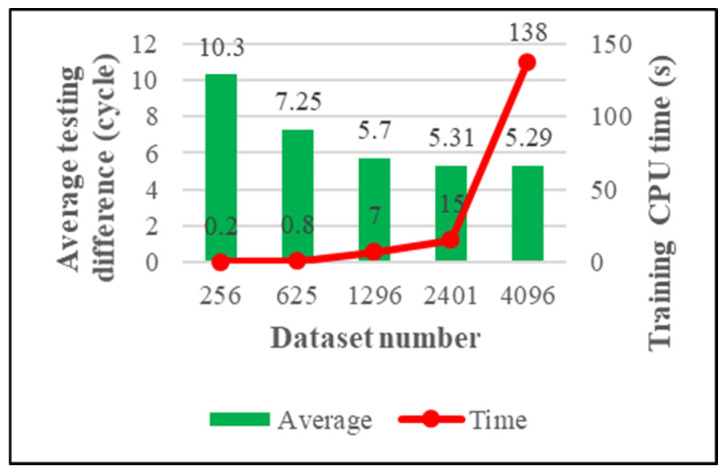
The average testing difference of the KRR model.

**Figure 10 materials-15-03897-f010:**
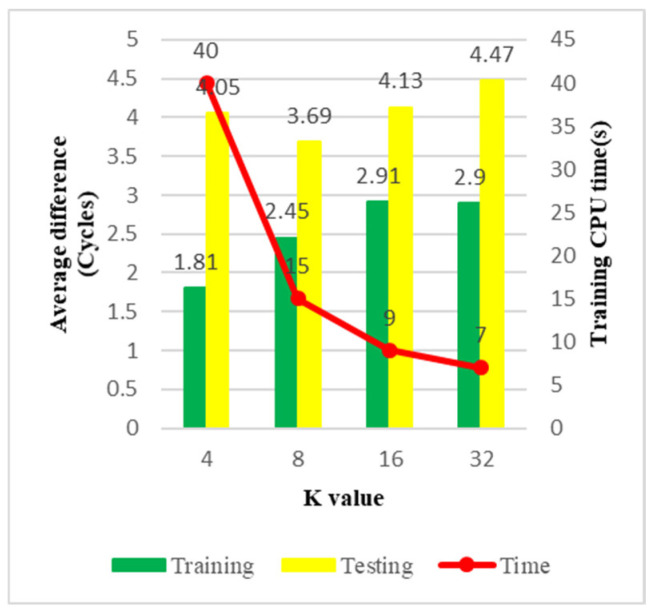
The average difference of the K-K model.

**Table 1 materials-15-03897-t001:** Dimension of WLP test vehicles [17,18].

	WLP-1	WLP-2	WLP-3	WLP-4	WLP-5
Silicon Chip	5.3×0.33 (mm)	4.0×0.33 (mm)	4.0×0.33 (mm)	4.0×0.33 (mm)	6.0×0.33 (mm)
Solder Ball Diameter	250 μm	250 μm	180 μm	200 μm	250 μm
Pitch	400 μm	400 μm	300 μm	300 μm	400 μm
Number of Solder Ball	121	100	144	144	196
MTTF (Cycles)	318	1013	587	876	904

**Table 2 materials-15-03897-t002:** Dimensions of fan-out WLP [19].

	FO-WLP
Packaging Size	14 mm×14 mm×0.1 mm
Chip Size	10 mm×10 mm×0.1 mm
Molding Compound Thickness	190 μm
Die-attach Film Thickness	10 mm×10 mm×0.01 mm
Solder Ball Diameter	250 μm
Pitch	400 μm
Number of Solder Ball	540
MTTF (Cycles)	249

**Table 3 materials-15-03897-t003:** Material properties for WLP [17,18].

Material	E (GPa)	υ	CTE (ppm/°C)
Solder Ball	Figure 5	0.35	25
Silicon Chip	150	0.28	2.62
Copper	68.9	0.34	16.7
SBL	2	0.33	55
Low-k	10	0.16	5
Solder Mask	6.87	0.35	19

**Table 4 materials-15-03897-t004:** Parameters for the Chaboche model.

T (K)	*σ* _0_	*C*	*γ*
233	47.64	8894.8	639.2
253	38.87	8573.3	660.0
313	24.06	6011.4	625.3
353	18.12	5804.2	697.7
395	14.31	4804.6	699.9

**Table 5 materials-15-03897-t005:** Reliability comparison for six test vehicles.

Test Vehicles	Experiment Reliability Life (Cycles)	Simulation Reliability Life (Cycles)	Difference (Cycles)	Difference (%)
WLP-1	318	319	1	0.3
WLP-2	1013	982	31	3.1
WLP-3	587	571	16	2.7
WLP-4	876	804	72	8.2
WLP-5	904	880	24	2.6
FO-WLP	249	248	1	0.4

**Table 6 materials-15-03897-t006:** Design parameters for 256 training data.

Design Parameters	Parameter Values
UPD	0.18, 0.2, 0.22, 0.24 (mm)
LPD	0.18, 0.2, 0.22, 0.24 (mm)
CT	0.15, 0.25, 0.35, 0.45 (mm)
SBLT	5, 14.17, 23.33, 32.5 (μm)
Total Number of Training Data	256

**Table 7 materials-15-03897-t007:** Design parameters for 625 training data.

Design Parameter	Parameter Values
UPD	0.18, 0.195, 0.21, 0.225, 0.24 (mm)
LPD	0.18, 0.195, 0.21, 0.225, 0.24 (mm)
CT	0.15, 0.225, 0.300, 0.375, 0.45 (mm)
SBLT	5, 11.88, 18.75, 25.63, 32.5 (μm)
Total Number of Training Data	625

**Table 8 materials-15-03897-t008:** Design parameters for 1296 training data.

Design Parameter	Parameter Values
UPD	0.18, 0.192, 0.204, 0.216, 0.228, 0.24 (mm)
LPD	0.18, 0.192, 0.204, 0.216, 0.228, 0.24 (mm)
CT	0.15, 0.21, 0.27, 0.33, 0.39, 0.45 (mm)
SBLT	5, 10.5, 16, 21.5, 27, 32.5 (μm)
Total Number of Training Data	1296

**Table 9 materials-15-03897-t009:** Design parameters for 2401 training data.

Design Parameter	Parameter Values
UPD	0.18, 0.19, 0.2, 0.21, 0.22, 0.23, 0.24 (mm)
LPD	0.18, 0.19, 0.2, 0.21, 0.22, 0.23, 0.24 (mm)
CT	0.15, 0.2, 0.25, 0.3, 0.35, 0.4, 0.45 (mm)
SBLT	5, 9.58, 14.17, 18.75, 23.33, 27.92, 32.5 (μm)
Total Number of Training Data	2401

**Table 10 materials-15-03897-t010:** Design parameters for 4096 training data.

Design Parameter	Parameter Values
UPD	0.18, 0.189, 0.197, 0.206, 0.214, 0.223, 0.231, 0.24 (mm)
LPD	0.18, 0.189, 0.197, 0.206, 0.214, 0.223, 0.231, 0.24 (mm)
CT	0.15, 0.189, 0.197, 0.206, 0.214, 0.223, 0.231, 0.24 (mm)
SBLT	5, 9.58, 14.17, 18.75, 23.33, 27.92, 32.5 (μm)
Total Number of Training Data	4096

**Table 11 materials-15-03897-t011:** Design parameters for 1296 testing data.

Design Parameter	Parameter Values
UPD	0.184, 0.194, 0.205, 0.219, 0.226, 0.234 (mm)
LPD	0.184, 0.194, 0.205, 0.219, 0.226, 0.234 (mm)
CT	0.174, 0.221, 0.289, 0.341, 0.379, 0.426 (mm)
SBLT	7.25, 12.55, 17.95, 22.65, 27.35, 30.35 (μm)
Total Number of Testing Data	1296

**Table 12 materials-15-03897-t012:** The performance of the KRR model.

	Training Model	256 Training Data	625 Training Data	1296 Training Data	2401 Training Data	4096 Training Data
Item						
α		0	0.001	0.001	0.001	0.001
γ		1	1688	1969	2	2
Maximum training diff. (%)		0%	3.45%	3.88%	3.31%	3.28%
Average training diff. (cycles)		0	6.23	5.64	5.73	5.58
Maximum testing diff. (%)		3.14%	2.95%	2.26%	2.79%	2.41%
Average testing diff. (cycles)		10.30	7.25	5.70	5.31	5.29
Training CPU time (sec.)		0.2	0.8	7	15	138

**Table 13 materials-15-03897-t013:** The performance of the KRR model with 9000 data.

	Training Model	9000 Training Data with 601 Testing Data
Item		
α		0.001
γ		1.0
Maximum training difference (%)		2.49%
Average training difference (cycles)		2.33
Maximum testing difference (%)		2.30%
Average testing difference (cycles)		3.24
Training CPU time (sec.)		1340

**Table 14 materials-15-03897-t014:** The performance of the sub-models.

	K Number	1	2	3	4
Item					
n		2222	2225	2266	2287
m		145	153	147	156
Maximum training difference (cycles)		25	7	34	26
Average training difference (cycles)		1.88	0.87	2.43	2.03
Maximum testing difference (cycles)		36	4	38	44
Average testing difference (cycles)		4.68	0.98	5.65	4.98

**Table 15 materials-15-03897-t015:** The performance of the K-K model in K = 4.

	Training Model	9000 Training Data with 601 Testing Data
Item		
Maximum training difference (%)		2.10%
Average training difference (cycles)		1.81
Maximum testing difference (%)		3.72%
Average testing difference (cycles)		4.05

**Table 16 materials-15-03897-t016:** The performance of the K-K model.

	K	4	8	16	24	32
Item						
Maximum training difference (%)		2.10	2.45	3.73	3.75	45
Average training difference (cycles)		1.81	2.45	2.91	2.06	2.90
Maximum testing difference (%)		3.72	3.44	3.66	4.57	4.57
Average testing difference (cycles)		4.05	3.69	4.13	3.71	4.47
Training CPU time (sec.)		40	15	9	8	7

**Table 17 materials-15-03897-t017:** The comparison of machine learning algorithms.

	KRR	K-K	ANN	RF	SVR
Maximum training difference (%)	2.49%	4.00%	3.99%	2.61%	3.55%
Average training difference (cycles)	2.33	2.90	3.74	3.77	1.16
Maximum testing difference (%)	2.30%	4.57%	2.89%	6.06%	2.43%
Average testing difference (cycles)	3.24	4.47	4.18	10.24	2.51
Training CPU time (sec.)	1340	7	469	8	1799

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
