# Peer review of "Predicting Wafer-Level Package Reliability Life Using Mixed Supervised and Unsupervised Machine Learning Algorithms"

_materials, 2022, doi:10.3390/ma15113897_

Round 1

Reviewer 1 Report

This paper combines Finite Element Analysis with machine learning algorithms to predict the reliability life of electronic packaging. Five training datasets and one testing dataset for machine learning were validated. I have following suggestions/comments:

  1. Simulation database for machine learning were built by following the lab modeling experiences. The database refers to old literature. The latest reference is from 2008. In the past decade there has been significant improvements in wafer-level packaging including the commercialization of wide band-gap semiconductors. It will be better to include recent models for the comparison.
  2. Number of solder balls and mean-time-to-failure must be numbers. However in Table 1 and Table 2, units are mentioned in mm. 
  3. What is the purpose of showing three WLP test vehicles in Table 1 and two WLP test vehicles in another table?  
  4. What are the limiting conditions while deciding the value of K?

Reviewer 2 Report

Dear Authors,

thanks for nice paper about WLP reliability study using AI algorithms.

Till now people (including me ;-) ) usually used FEA modeling and simulation tools (like: Coventor, ANSYS, Comsol etc..) and/or thermal/accelerated physical tests what was time and hardware consuming.

Here Authors presented new approach - which is promising to get better accuracy and speed up the modeling and simulation time.

Generally, not too much comments from my side, paper is prepared in correct way, English is good, all sections from introduction to final conclusions has been written in proper way.

Reviewer 3 Report

This paper proposes the hybrid model with Kernel Ridge Regression (KRR) algorithm to  predicting the reliability life of electronic packaging. The proposed hybrid k-k model algorithm illustrates better computational efficiency and higher accuracy compared with other algorithms. There are two comments on the paper. In order to verify the precision and computational efficiency of the proposed algorithm, the algorithm with training data and different K values should be applied to other testing conditions such as other test vehicles. The value of K between 16 and 32 should be utilized to simulate the error(%) and cpu time, so that the relationship between accuracy and cpu time can be identified as analytical function.

Round 2

Reviewer 1 Report

Thank you for implementing the suggestions in the revised manuscript. Please make sure the legends of the respective figures/table should come with the figure/table and all the figures and tables should be completely visible to the readers.